# Gradient Perturbation is Underrated for Differentially Private Convex Optimization

## Abstract

Gradient perturbation, widely used for differentially private optimization, injects noise at every iterative update to guarantee differential privacy. Previous work first determines the noise level that can satisfy the privacy requirement and then analyzes the utility of noisy gradient updates as in non-private case. In this paper, we explore how the privacy noise affects the optimization property. We show that for differentially private convex optimization, the utility guarantee of both DP-GD and DP-SGD is determined by an *expected curvature* rather than the minimum curvature. The *expected curvature* represents the average curvature over the optimization path, which is usually much larger than the minimum curvature and hence can help us achieve a significantly improved utility guarantee. By using the *expected curvature*, our theory justifies the advantage of gradient perturbation over other perturbation methods and closes the gap between theory and practice. Extensive experiments on real world datasets corroborate our theoretical findings.

## 1 Introduction

Machine learning has become a powerful tool for many practical applications. The training process often needs access to some private dataset, e.g., applications in financial and medical fields. Recent work has shown that the model learned from training data may leak unintended information of individual records (Fredrikson et al., 2015; Wu et al., 2016; Shokri et al., 2017; Hitaj et al., 2017). It is known that *Differential privacy (DP)* (Dwork et al., 2006a;b) is a golden standard for privacy preserving data analysis. It provides provable privacy guarantee by ensuring the influence of any individual record is negligible. It has been deployed into real world applications by large-scale corporations and U.S. Census Bureau (Erlingsson et al., 2014; McMillan, 2016; Abowd, 2016; Ding et al., 2017).

We study the fundamental problem when differential privacy meets machine learning: the *differentially private empirical risk minimization (DP-ERM)* problem (Chaudhuri & Monteleoni, 2009; Chaudhuri et al., 2011; Kifer et al., 2012; Bassily et al., 2014; Talwar et al., 2015; Wu et al., 2017; Zhang et al., 2017; Wang et al., 2017; Smith et al., 2017; Jayaraman et al., 2018; Feldman et al., 2018; Iyengar et al., 2019; Wang & Gu, 2019). DP-ERM minimizes the empirical risk while guaranteeing that the output of learning algorithm is differentially private with respect to the training data. Such privacy guarantee provides strong protection against potential adversaries (Hitaj et al., 2017; Rahman et al., 2018). In order to guarantee privacy, it is necessary to introduce randomness to the algorithm. There are usually three ways to introduce randomness according to the time of adding noise: *output perturbation*, *objective perturbation* and *gradient perturbation*.

*Output perturbation* (Wu et al., 2017; Zhang et al., 2017) first runs the learning algorithm the same as in the non-private case then adds noise to the output parameter. *Objective perturbation* (Chaudhuri et al., 2011; Kifer et al., 2012; Iyengar et al., 2019) perturbs the objective (i.e., the empirical loss) then release the minimizer of the perturbed objective. *Gradient perturbation* (Song et al., 2013; Bassily et al., 2014; Abadi et al., 2016; Wang et al., 2017; Lee & Kifer, 2018; Jayaraman et al., 2018) perturbs each intermediate update. If each

update is differentially private, the composition theorem of differential privacy ensures the whole learning procedure is differentially private.

Gradient perturbation comes with several advantages over output/objective perturbations. Firstly, gradient perturbation does not require strong assumption on the objective because it only needs to bound the sensitivity of gradient update rather than the whole learning process. Secondly, gradient perturbation can release the noisy gradient at each iteration without damaging the privacy guarantee as differential privacy is immune to *post processing* (Dwork et al., 2014). Thus, it is a more favorable choice for certain applications such as distributed optimization (Rajkumar & Agarwal, 2012; Agarwal et al., 2018; Jayaraman et al., 2018). At last, gradient perturbation often achieves better empirical utility than output/objective perturbations for DP-ERM.

However, the existing theoretical utility guarantee for gradient perturbation is the same as or strictly inferior to that of other perturbation methods as shown in Table 1. This motivates us to ask

"What is wrong with the theory for gradient perturbation? Can we justify the empirical advantage of gradient perturbation theoretically?"

We revisit the analysis for gradient perturbation approach. Previous work (Bassily et al., 2014; Wang et al., 2017; Jayaraman et al., 2018) derive the utility guarantee of gradient perturbation via two steps. They first determine the noise variance at each step that meets the privacy requirement and then derive the utility guarantee by using the convergence analysis the same as in non-private case. However, the noise to guarantee privacy naturally affects the optimization procedure, but previous approach does not exploit the interaction between privacy noise and optimization of gradient perturbation.

In this paper, we utilize the fact the privacy noise affects the optimization procedure and establish new and much tighter utility guarantees for gradient perturbation approaches. Our contribution can be summarized as follows.

- We introduce an *expected curvature* that can characterize the optimization property accurately when there is perturbation noise at each gradient update.
- We establish the utility guarantees for DP-GD for both convex and strongly convex objectives based on the *expected curvature* rather than the usual minimum curvature.
- We also establish the the utility guarantees for DP-SGD for both convex and strongly convex objectives based on the *expected curvature*. To the best of our knowledge, this is the first work to remove the dependency on minimum curvature for DP-ERM algorithms.

In DP-ERM literature, there is a gap between the utility guarantee of non-strongly convex objectives and that of strongly convex objectives. However, by using the *expected curvature*, we show that some of the non-strongly convex objectives can achieve the same order of utility guarantee as the strongly convex objectives, matching the empirical observation. This is because the expected curvature could be relatively large even for non-strongly convex objectives.

As we mentioned earlier, prior to our work, there is a mismatch between theoretical guarantee and empirical observation of gradient perturbation approach compared with other two perturbation approaches. Our result theoretically justifies the advantage of gradient perturbation and close the mismatch.

## 1.1 Paper Organization

The rest of this paper is organized as follows. Section 2 introduces notations and the DP-ERM task. In Sections 3, we first introduce the expected curvature and establish the utility guarantee of both DP-GD and DP-SGD based on such expected curvature. Then we give some discussion on three perturbation approaches. We conduct extensive experiments in Section 4. Finally, we conclude in Section 5.

Table 1: Expected excess empirical risk bounds under $(\epsilon, \delta)$-DP, where $n$ and $p$ are the number of samples and the number of parameters, respectively, and $\beta, \mu$ and $\nu$ are the smooth coefficient, the strongly convex coefficient and the *expected curvature*, respectively, and $\nu \geq \mu$ (see Section 3.1). We note that $\mu = 0$ denotes the convex but not strongly convex objective. The Lipschitz constant $L$ is assumed to be 1. We omit $\log(1/\delta)$ for simplicity.

| Authors | Perturbation | Algorithm | Utility ($\mu = 0$) | Utility ($\mu > 0$) |
|---|---|---|---|---|
| Chaudhuri et al. (2011) | Objective | N/A | $\mathcal{O}\left(\frac{\sqrt{p}}{n\epsilon}\right)$ | $\mathcal{O}\left(\frac{p}{\mu n^2 \epsilon^2}\right)$ |
| Zhang et al. (2017) | Output | GD | $\mathcal{O}\left((\frac{\sqrt{\beta p}}{n\epsilon})^{2/3}\right)$ | $\mathcal{O}\left(\frac{\beta p}{\mu^2 n^2 \epsilon^2}\right)$ |
| Bassily et al. (2014) | Gradient | SGD | $\mathcal{O}\left(\frac{\sqrt{p}\log^{3/2}(n)}{n\epsilon}\right)$ | $\mathcal{O}\left(\frac{p\log^2(n)}{\mu n^2 \epsilon^2}\right)$ |
| Jayaraman et al. (2018) | Gradient | GD | N/A | $\mathcal{O}\left(\frac{\beta p\log^2(n)}{\mu^2 n^2 \epsilon^2}\right)$ |
| Ours | Gradient | GD | $\mathcal{O}\left(\frac{\sqrt{p}}{n\epsilon} \wedge \frac{\beta p\log(n)}{\nu^2 n^2 \epsilon^2}\right)$ | $\mathcal{O}\left(\frac{\beta p\log(n)}{\nu^2 n^2 \epsilon^2}\right)$ |
| Ours | Gradient | SGD | $\mathcal{O}\left(\frac{\sqrt{p}\log(n)}{n\epsilon} \wedge \frac{p\log(n)}{\nu n^2 \epsilon^2}\right)$ | $\mathcal{O}\left(\frac{p\log(n)}{\nu n^2 \epsilon^2}\right)$ |

## 2 PRELIMINARY

We introduce notations and definitions in this section. Given dataset $D = \{d_1, \ldots, d_n\}$, the objective function $F(\boldsymbol{x}; D)$ is defined as $F(\boldsymbol{x}; D) \triangleq \frac{1}{n}\sum_{i=1}^{n} f(\boldsymbol{x}; d_i)$, where $f(\boldsymbol{x}; d_i) : \mathbb{R}^p \to \mathbb{R}$ is the loss of model $\boldsymbol{x} \in \mathbb{R}^p$ for the record $d_i$.

For simplicity, we use $F(\boldsymbol{x})$ to denote $F(\boldsymbol{x}; D)$. We use $\|\boldsymbol{v}\|$ to denote the $l_2$ norm of a vector $\boldsymbol{v}$. We use $\mathcal{X}_f^* = \arg\min_{\boldsymbol{x} \in \mathbb{R}^p} f(\boldsymbol{x})$ to denote the set of optimal solutions of $f(\boldsymbol{x})$. Throughout this paper, we assume $\mathcal{X}_f^*$ non-empty.

**Definition 1** (Objective properties). *For any $\boldsymbol{x}, \boldsymbol{y} \in \mathbb{R}^p$, a function $f : \mathbb{R}^p \to \mathbb{R}$*

- *is $L$-Lipschitz if $|f(\boldsymbol{x}) - f(\boldsymbol{y})| \leq L \|\boldsymbol{x} - \boldsymbol{y}\|$.*

- *is $\beta$-smooth if $f(\boldsymbol{y}) \leq f(\boldsymbol{x}) + \langle \nabla f(\boldsymbol{x}), \boldsymbol{y} - \boldsymbol{x} \rangle + \frac{\beta}{2}\|\boldsymbol{y} - \boldsymbol{x}\|^2$.*

- *is convex if $\langle \nabla f(\boldsymbol{x}) - \nabla f(\boldsymbol{y}), \boldsymbol{x} - \boldsymbol{y} \rangle \geq 0$.*

- *is $\mu$-strongly convex (or $\mu$-SC) if $\langle \nabla f(\boldsymbol{x}) - \nabla f(\boldsymbol{y}), \boldsymbol{x} - \boldsymbol{y} \rangle \geq \mu \|\boldsymbol{x} - \boldsymbol{y}\|^2$.*

The strong convexity coefficient $\mu$ is the lower bound of the minimum curvature of function $f$ over the domain.

We say that two datasets $D, D'$ are neighboring datasets (denoted as $D \sim D'$) if $D$ can be obtained by arbitrarily modifying one record in $D'$ (or vice versa). In this paper we consider $(\epsilon, \delta)$-differential privacy as follows.

**Definition 2** ( $(\epsilon, \delta)$-DP (Dwork et al., 2006a;b)). *A randomized mechanism $\mathcal{M} : D \to \mathcal{R}$ guarantees $(\epsilon, \delta)$-differential privacy if for any two neighboring input datasets $D, D'$ and for any subset of outputs $S \subseteq \mathcal{R}$ it holds that $Pr[\mathcal{M}(D) \in S] \leq e^\epsilon Pr[\mathcal{M}(D') \in S] + \delta$.*

We note that $\delta$ can be viewed as the probability that original $\epsilon$-DP fails and a meaningful setting requires $\delta \ll \frac{1}{n}$. By its definition, differential privacy controls the maximum influence that any individual record can produce. Smaller $\epsilon, \delta$ implies less information leak but usually leads to worse utility. One can adjust $\epsilon, \delta$ to trade off between privacy and utility.

DP-ERM requires the output $\boldsymbol{x}_{out} \in \mathbb{R}^p$ is differentially private with respect to the input dataset $D$. Let $\boldsymbol{x}_* \in \mathcal{X}_F^*$ be one of the optimal solutions of $F(\boldsymbol{x})$, the utility of DP-ERM algorithm is measured by *expected excess empirical risk*: $\mathbb{E}[F(\boldsymbol{x}_{out}) - F(\boldsymbol{x}_*)]$, where the expectation is taken over the algorithm randomness.

## 3 Main Results

In this section, we first define the *expected curvature* $\nu$ and explain why it depends only on the average curvature. We then use such expected curvature to improve the analysis of both DP-SGD and DP-GD.

### 3.1 Expected Curvature

In non-private setting, the analysis of convex optimization relies on the strongly convex coefficient $\mu$, which is the minimum curvature over the domain and can be extremely small for some common objectives. Previous work on DP-ERM uses the same analysis as in non-private case and therefore the resulting utility bounds rely on the minimum curvature. In our analysis, however, we avoid the dependency on the minimum curvature by exploiting how the privacy noise affects the optimization. With the perturbation noise, the expected curvature that the optimization path encounters is related to the average curvature instead of the minimum curvature. Definition 3 uses $\nu$ to capture such average curvature with Gaussian noise. We use $\boldsymbol{x}_* = \arg\min_{\boldsymbol{x} \in \mathcal{X}_*} \|\boldsymbol{x} - \boldsymbol{x}_1\|$ to denote the closest solution to the initial point.

**Definition 3** (Expected curvature). *A convex function $F : \mathbb{R}^p \to \mathbb{R}$, has* expected curvature *$\nu$ with respect to noise $\mathcal{N}(0, \sigma^2 \boldsymbol{I}_p)$ if for any $\boldsymbol{x} \in \mathbb{R}^p$ and $\tilde{\boldsymbol{x}} = \boldsymbol{x} - \boldsymbol{z}$ where $\boldsymbol{z} \sim \mathcal{N}(0, \sigma^2 \boldsymbol{I}_p)$, it holds that*

$$\mathbb{E}[\langle \nabla F(\tilde{\boldsymbol{x}}), \tilde{\boldsymbol{x}} - \boldsymbol{x}_* \rangle] \geq \nu \mathbb{E}[\|\tilde{\boldsymbol{x}} - \boldsymbol{x}_*\|^2], \tag{1}$$

*where the expectation is taken with respect to $\boldsymbol{z}$.*

**Claim 1.** *If $F$ is $\mu$-strongly convex, we have $\nu \geq \mu$.*

*Proof.* It can be verified that $\nu = \mu$ always holds because of the strongly convex definition. $\square$

**In fact, $\nu$ represents the average curvature and is much larger than $\mu$.** We use $\boldsymbol{x}'$ to denote the transpose of $\boldsymbol{x}$. Let $\boldsymbol{H_x} = \nabla^2 F(\boldsymbol{x})$ be the Hessian matrix evaluated at $\boldsymbol{x}$. We use Taylor expansion to approximate the left hand side of Eq (1) as follows

$$\begin{aligned} \mathbb{E}[\langle \nabla F(\tilde{\boldsymbol{x}}), \tilde{\boldsymbol{x}} - \boldsymbol{x}_* \rangle] &\approx \mathbb{E}[\langle \nabla F(\boldsymbol{x}) - \boldsymbol{H_x z}, \boldsymbol{x} - \boldsymbol{z} - \boldsymbol{x}_* \rangle] \\ &= \langle \nabla F(\boldsymbol{x}), \boldsymbol{x} - \boldsymbol{x}_* \rangle + \mathbb{E}[\boldsymbol{z}' H_{\boldsymbol{x}} \boldsymbol{z}] \\ &= \langle \nabla F(\boldsymbol{x}), \boldsymbol{x} - \boldsymbol{x}_* \rangle + \sigma^2 \operatorname{tr}(\boldsymbol{H_x}). \end{aligned}$$

For convex objective, the Hessian matrix is positive semi-definite and $\operatorname{tr}(\boldsymbol{H_x})$ is the sum of the eigenvalues of $H_{\boldsymbol{x}}$. We can further express out the right hand side of Eq (1) as follows

$$\mathbb{E}[\|\tilde{\boldsymbol{x}} - \boldsymbol{x}_*\|^2] = \mathbb{E}[\|\boldsymbol{x} - \boldsymbol{z} - \boldsymbol{x}_*\|^2] = \nu \left( \|\boldsymbol{x} - \boldsymbol{x}_*\|^2 + p\sigma^2 \right).$$

Based on the above approximation, we can estimate the value of $\nu$ in Definition 3: $\nu \lesssim \frac{\operatorname{tr}(\boldsymbol{H_x})\sigma^2 + \mu\|\boldsymbol{x} - \boldsymbol{x}_*\|^2}{p\sigma^2 + \|\boldsymbol{x} - \boldsymbol{x}_*\|^2}$. For relatively large $\sigma^2$, this implies $\nu \approx \frac{\operatorname{tr}(\boldsymbol{H_x})}{p}$ that is the average curvature at $\boldsymbol{x}$. Large variance is a reasonable setting because meaningful differential privacy guarantee requires non-trivial amount of noise.

The above analysis suggests that $\nu$ can be independent of and much larger than $\mu$. This is indeed true for many convex objectives. Let us take the $l_2$ regularized logistic regression as an example. The objective is strongly convex only due to the $l_2$ regularizer. Thus, the minimum curvature (strongly convex coefficient) is the regularization coefficient $\lambda$. Sharmir et al. [1] shows the optimal choice of $\lambda$ is $\Theta(n^{-1/2})$ (Section 4.3 in [1]). In practice, typical choice of $\lambda$ is even smaller and could be on the order of $n^{-1}$. Figure 1 compares the minimum and average curvatures of regularized logistic regression during the training process. The average curvature is basically unaffected by the regularization term $\lambda$. In contrast, the minimum curvature reaches $\lambda$ in first few steps. Therefore removing the dependence on minimum curvature is a significant improvement. We also plot the curvatures for another dataset KDDCup99 in the Appendix C. The resulting curvatures are similar to Figure 1.

**Perturbation noise is necessary to attain $\nu > \mu$.** We note that $\nu = \mu$ when the training process does not involve perturbation noise (corresponding to $\sigma = 0$ in Definition 3). For

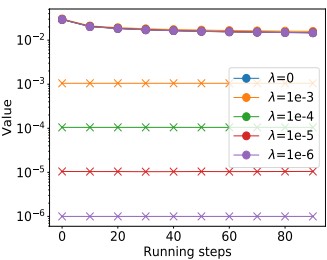

Figure 1: Curvatures of regularized logistic regression on Adult dataset over training. Dot/cross symbol represents average/minimum curvature respectively.

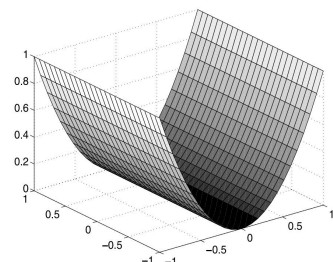

Figure 2: Illustration of a generic loss function in the high dimensional setting ($p > n$, Figure 3 in Negahban et al. (2012)).

example, objective/output perturbation cannot utilize this expected curvature condition as no noise is injected in their training process. Therefore, among three existing perturbation methods, gradient perturbation is the only method can leverage such effect of noise.

**We note that $\mu = 0$ does not necessarily lead to $\nu = 0$.** A concrete example is given in Figure 2 (from Negahban et al. (2012)). It provides an illustration of the loss function in the high-dimensional ($p > n$) setting, i.e., the resticted strongly convex scenario: the loss is curved in certain directions but completely flat in others. The average curvature of such objective is always positive but the worst curvature is 0. Though some recent work shows the utility guarantee of high dimensional DP-ERM task may not depend on the worst curvature (Wang & Gu, 2019), Figure 2 still provides a good illustration for the case of $\nu > \mu = 0$. Moreover, as shown in Figure 1, the average curvature of logistic regression on Adult dataset is above 0 during the training procedure even the regularization term is 0. As we will show later, a positive $\nu$ over the optimization path is sufficient for our optimization analysis.

### 3.2 Utility Guarantee of DP-GD Based on Expected Curvature

In this section we show that the *expected curvature* can be used to improve the utility bound of DP-GD (Algorithm 1).

---
**Algorithm 1:** Differentially Private Gradient Descent (DP-GD)

---
**Input:** Privacy parameters $\epsilon, \delta$; running steps $T$; learning rate $\eta$. Loss function $F(\boldsymbol{x})$ with Lipschitz constant $L$.
**for** $t = 1$ **to** $T$ **do**
    Compute $\boldsymbol{g}_t = \nabla F(\boldsymbol{x}_t)$.
    Update parameter $\boldsymbol{x}_{t+1} = \boldsymbol{x}_t - \eta_t(\boldsymbol{g}_t + \boldsymbol{z}_t)$, where $\boldsymbol{z}_t \sim \mathcal{N}(0, \sigma_t^2 I_p)$.
**end for**

---

Algorithm 1 is $(\epsilon, \delta)$-DP if we set $\sigma_t = \Theta\left(\frac{L\sqrt{T\log(1/\delta)}}{n\epsilon}\right)$ (Jayaraman et al., 2018). Let $\boldsymbol{x}_1, \ldots, \boldsymbol{x}_T$ be the training path and $\nu = \min\{\nu_1, \ldots, \nu_T\}$ be the minimum expected curvature over the path. Now we present the utility guarantee of DP-GD for the case of $\nu > 0$.

**Theorem 1** (Utility guarantee, $\nu > 0$.)**.** *Suppose $F$ is $L$-Lipschitz and $\beta$-smooth with $\nu$ expected curvature. Set $\eta \leq \frac{1}{\beta}$, $T = \frac{2\log(n)}{\eta\nu}$ and $\sigma_t = \Theta\left(L\sqrt{T\log(1/\delta)}/n\epsilon\right)$, we have*

$$\mathbb{E}[F(\boldsymbol{x}_{T+1}) - F(\boldsymbol{x}_*)] = \mathcal{O}\left(\frac{\beta p \log(n) L^2 \log(1/\delta)}{\nu^2 n^2 \epsilon^2}\right).$$

*Proof.* All proofs in this paper are relegated to Appendix A. □

**Remark 1.** *Theorem 3 only depends on the expected curvature over the training path $\nu$.*

The expectation is taken over the algorithm randomness if without specification. Theorem 1 significantly improves the original analysis of DP-GD because of our arguments in Section 3.1. If $\nu = 0$, then the curvatures are flatten in all directions. One example is the linear function, which is used by Bassily et al. (2014) to derive their utility lower bound. Such simple function may not be commonly used as loss function in practice. Nonetheless, we give the utility guarantee for the case of $\nu = 0$ in Theorem 2.

**Theorem 2** (Utility guarantee, $\nu = 0$.). *Suppose $F$ is $L$-Lipschitz and $\beta$-smooth. Set $\eta = \frac{1}{\beta}$, $T = \frac{n\beta\epsilon}{\sqrt{p}}$ and $\sigma_t = \Theta\left(L\sqrt{T\log(1/\delta)}/n\epsilon\right)$. Let $\bar{\boldsymbol{x}} = \frac{1}{T}\sum_{i=1}^{T}\boldsymbol{x}_{i+1}$, we have*

$$\mathbb{E}[F(\bar{\boldsymbol{x}}) - F(\boldsymbol{x}_*)] = \mathcal{O}\left(\frac{\sqrt{p}L^2\log(1/\delta)}{n\epsilon}\right).$$

We use parameter averaging to reduce the influence of perturbation noise because gradient update does not have strong contraction effect when $\nu = 0$.

### 3.3 Utiltiy Guarantee of DP-SGD Based on Expected Curvature

Stochastic gradient descent has become one of the most popular optimization methods because of the cheap one-iteration cost. In this section we show that *expected curvature* can also improve the utility analysis for DP-SGD (Algorithm 2). We note that $\nabla f(\boldsymbol{x})$ represents an element from the subgradient set evaluated at $\boldsymbol{x}$ when the objective is not smooth. Before stating our theorem, we introduce the *moments accountant* technique (Lemma 1) that is essential to establish privacy guarantee.

**Lemma 1** (Abadi et al. (2016)). *There exist constants $c_1$ and $c_2$ so that given running steps $T$, for any $\epsilon < c_1 T/n^2$, Algorithm 2 is $(\epsilon, \delta)$-differentially private for any $\delta > 0$ if we choose $\sigma \geq c_2 \frac{\sqrt{T log(1/\delta)}}{n\epsilon}$.*

---

**Algorithm 2:** Differentially Private Stochastic Gradient Descent (DP-SGD)

**Input** : Dataset $D = \{d_1, \ldots, d_n\}$. Individual loss function: $f_i(\boldsymbol{x}) = f(\boldsymbol{x}; d_i)$ with Lipschitz constant $L$. Number of iterations: $T$. Learning rate: $\eta_t$.

1 **for** $t = 1$ *to* $T$ **do**
2     Sample $i_t$ from $[n]$ uniformly.
3     Compute $\boldsymbol{g}_t = \nabla f_{i_t}(\boldsymbol{x}_t)$.
4     Update parameter $\boldsymbol{x}_{t+1} = \boldsymbol{x}_t - \eta_t(\boldsymbol{g}_t + \boldsymbol{z}_t)$, where $\boldsymbol{z}_t \sim \mathcal{N}\left(0, L^2\sigma^2 I_p\right)$.
5 **end**

---

For the case of $\nu > 0$, Theorem 3 presents the utility guarantee of DP-SGD.

**Theorem 3** (Utility guarantee, $\nu > 0$.). *Suppose $F$ is $L$-Lipschitz with $\nu$ expected curvature. Choose $\sigma$ based on Lemma 1 to guarantee $(\epsilon, \delta)$-DP. Set $\eta_t = \frac{1}{\nu t}$ and $T = n^2\epsilon^2$, we have*

$$\mathbb{E}[F(\boldsymbol{x}_T) - F(\boldsymbol{x}_*)] = \mathcal{O}\left(\frac{pL^2\log(n)\log(1/\delta)}{n^2\epsilon^2\nu}\right).$$

**Remark 2.** *Theorem 3 does not require smooth assumption.*

Theorem 3 shows the utility guarantee of DP-SGD also depends on $\nu$ rather than $\mu$. We set $T = \Theta(n^2)$ following Bassily et al. (2014). We note that $T = \Theta(n^2)$ is necessary even for non-private SGD to reach $1/n^2$ precision. We next show for a relatively coarse precision, the running time can be reduced significantly.

**Theorem 4.** *Suppose $F$ is $L$-Lipschitz with $\nu$ expected curvature. Choose $\sigma$ based on Lemma 1 to guarantee $(\epsilon, \delta)$-DP. Set $\eta_t = \frac{1}{\nu t}$ and $T = \frac{n\epsilon}{\sqrt{p}}$. Suppose $p < n^2$, we have*

$$\mathbb{E}[F(\boldsymbol{x}_T) - F(\boldsymbol{x}_*)] = \mathcal{O}\left(\frac{\sqrt{p}L^2\log(n)}{n\epsilon\nu}\right).$$

We note that the analysis of Bassily et al. (2014) yields $\mathbb{E}[F(\boldsymbol{x}_T) - F(\boldsymbol{x}_*)] = \mathcal{O}\left(\frac{\sqrt{p}L^2 \log^2(n)}{n\epsilon\mu}\right)$ if setting $T = \frac{n\epsilon}{\sqrt{p}}$, which still depends on the minimum curvature. Theorem 5 shows the utility for the case of $\nu = 0$.

**Theorem 5** (Utility guarantee, $\nu = 0$.). *Suppose $F$ is $L$-Lipschitz. Assume $\|\boldsymbol{x}_t\| \leq D$ for $t \in [T]$. Choose $\sigma$ based on Lemma 1 to guarantee $(\epsilon, \delta)$-DP. Let $G = L\sqrt{1 + p\sigma^2}$, set $\eta_t = \frac{D}{G\sqrt{t}}$ and $T = n^2\epsilon^2$, we have*

$$\mathbb{E}[F(\boldsymbol{x}_T) - F(\boldsymbol{x}_*)] = \mathcal{O}\left(\frac{\sqrt{p \log(1/\delta)} L \log(n)}{n\epsilon}\right).$$

This utility guarantee can be derived from Theorem 2 in (Shamir & Zhang, 2013).

### 3.4 Discussion on three perturbation approaches.

In this section, we briefly discuss two other perturbation approaches and compare them to the gradient perturbation approach.

*Output perturbation* (Wu et al., 2017; Zhang et al., 2017) perturbs the learning algorithm after training. It adds noise to the resulting model of non-private learning process. The magnitude of perturbation noise is propositional to the maximum influence one record can cause on the learned model. Take the gradient descent algorithm as an example. At each step, the gradient of different records would diverge the two sets of parameters generated by neighboring datasets, the maximum distance expansion is related to the Lipschitz coefficient. At the same time, the gradient of the same records in two datasets would shrink the parameter distance because of the contraction effect of the gradient update. The contraction effect depends on the smooth and strongly convex coefficient. Smaller strongly convex coefficient leads to weaker contraction. The sensitivity of output perturbation algorithm is the upper bound on the largest possible final distance between two sets of parameters.

*Objective perturbation* (Chaudhuri et al., 2011; Kifer et al., 2012; Iyengar et al., 2019) perturbs the objective function before training. It requires the objective function to be strongly convex to guarantee the uniqueness of the solution. It first adds $L_2$ regularization to obtain strong convexity if the original objective is not strongly convex. Then it perturbs the objective with a random linear term. The sensitivity of objective perturbation is the maximum change of the minimizer that one record can produce. Chaudhuri et al. (2011) and Kifer et al. (2012) use the largest and the smallest eigenvalue (i.e. the smooth and strongly convex coefficient) of the objective's Hessian matrix to upper bound such change.

In comparison, gradient perturbation is more flexible than output/objective perturbation. For example, to bound the sensitivity, gradient perturbation only requires Lipschitz coefficient which can be easily obtained by using the gradient clipping technique. However, both output and objective perturbation further need to compute the smooth coefficient, which is hard for some common objectives such as softmax regression.

More critically, output/objective perturbation cannot utilize the expected curvature condition because their training process does not contain perturbation noise. Moreover, they have to consider the worst performance of learning algorithm. That is because DP makes the worst case assumption on query function and output/objective perturbation treat the whole learning algorithm as a single query to private dataset. This explains why their utility guarantee depends on the worst curvature of the objective.

## 4 Experiment

In this section, we evaluate the performance of DP-GD and DP-SGD on multiple real world datasets. We use the benchmark datasets provided by Iyengar et al. (2019). Objective functions are *logistic regression* and *softmax regression* for binary and multi-class datasets, respectively.

**Datasets.** The benchmark datasets includes two multi-class datasets (MNIST, Covertype) and five binary datasets, and three of them are high dimensional (Gisette, Real-sim, RCV1).

Table 2: Algorithm validation accuracy (in %) on various kinds of real world datasets. Privacy parameter $\epsilon$ is 0.1 for binary dataset and 1 for multi-classes datasets.

|             | KDDCup99 | Adult | MNIST | Covertype | Gisette | Real-sim | RCV1 |
|-------------|----------|-------|-------|-----------|---------|----------|------|
| Non private | 99.1     | 84.8  | 91.9  | 71.2      | 96.6    | 93.3     | 93.5 |
| AMP[1]      | 97.5     | 79.3  | 71.9  | 64.3      | 62.8    | 73.1     | 64.5 |
| Out-SGD     | 98.1     | 77.4  | 69.4  | 62.4      | 62.3    | 73.2     | 66.7 |
| DP-SGD      | 98.7     | 80.4  | 87.5  | **67.7**  | 63.0    | 73.8     | 70.4 |
| DP-GD       | **98.7** | **80.9** | **88.6** | 66.2  | **67.3** | **76.1** | **74.9** |

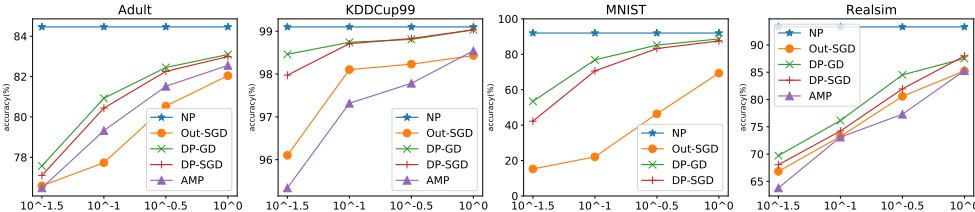

Figure 3: Algorithm validation accuracy (in %) with varying $\epsilon$. NP represents non-private baseline. Detailed description about evaluated datasets can be found in Table 3.

Following Iyengar et al. (2019), we use 80% data for training and the rest for testing. Detailed description of datasets can be found in Appendix B

**Implementation details.** We track *Rényi differentialy privacy (RDP) (Mironov, 2017)* and convert it to $(\epsilon, \delta)$-DP. Running step $T$ is chosen from $\{50, 200, 800\}$ for both DP-GD and DP-SGD. For DP-SGD, we use moments accountant to track the privacy loss and the sampling ratio is set as 0.1. The standard deviation of the added noise $\sigma$ is set to be the smallest value such that the privacy budget is allowable to run desired steps. We ensure each loss function is Lipschitz by clipping individual gradient. The method in Goodfellow (2015) allows us to clip individual gradient efficiently. Clipping threshold is set as 1 (0.5 for high dimensional datasets because of the sparse gradient). For DP-GD, learning rate is chosen from $\{0.1, 1.0, 5.0\}$ ($\{0.2, 2.0, 10.0\}$ for high dimensional datasets). The learning rate of DP-SGD is twice as large as DP-GD and it is divided by 2 at the middle of training. Privacy parameter $\delta$ is set as $\frac{1}{n^2}$. The $l_2$ regularization coefficient is set as $1 \times 10^{-4}$. All reported numbers are averaged over 20 runs.

**Baseline algorithms.** The baseline algorithms include state-of-the-art objective and output perturbation algorithms. For objective perturbation, we use *Approximate Minima Perturbation (AMP)* (Iyengar et al., 2019). For output perturbation, we use the algorithm in Wu et al. (2017) (Output perturbation SGD). We adopt the implementation and hyperparameters in Iyengar et al. (2019) for both algorithms. For multi-class classification tasks, Wu et al. (2017) and Iyengar et al. (2019) divide the privacy budget evenly and train multiple binary classifiers because their algorithms need to compute smooth coefficient before training and therefore are not directly applicable to softmax regression.

**Experiment results.** The validation accuracy results for all evaluated algorithms with $\epsilon = 0.1$ (1.0 for multi-class datasets) are presented in Table 2. We also plot the accuracy results with varying $\epsilon$ in Figure 3. These results confirm our theory in Section 3: gradient perturbation achieves better performance than other perturbation methods as it leverages the average curvature.

## 5 CONCLUSION

In this paper, we show the privacy noise actually helps optimization analysis, which can be used to improve the utility guarantee of both DP-GD and DP-SGD. Our result theoretically

---

[1]For multi-class datas sets MNIST and Covertype, we use the numbers reported in Iyengar et al. (2019) directly because of the long running time of AMP especially on multi-class datasets.

justifies the empirical superiority of gradient perturbation over other methods and advance the state of the art utility guarantee of DP-ERM algorithms. Experiments on real world datasets corroborate our theoretical findings nicely. In the future, it is interesting to consider how to utilize the expected curvature condition to improve the utility guarantee of other gradient perturbation based algorithms.

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

## APPENDIX A   PROOFS RELATED TO DP-GD AND DP-SGD

*Proof of Theorem 1.* Let $\boldsymbol{x}_1, \ldots, \boldsymbol{x}_t$ be the path generated by optimization procedure. Since $\boldsymbol{x}_t$ contains Gaussian perturbation noise $\boldsymbol{z}_{t-1}$, Definition 3 gives us

$$\mathbb{E}_{\boldsymbol{z}_{t-1}}[\langle \boldsymbol{x}_t - \boldsymbol{x}_*, \nabla F(\boldsymbol{x}_t)\rangle] \geq \nu_t \mathbb{E}_{\boldsymbol{z}_{t-1}}[\|\boldsymbol{x}_t - \boldsymbol{x}_*\|^2].$$

Since $F$ is $\beta$-smooth, we have

$$\langle \boldsymbol{x}_t - \boldsymbol{x}_*, \nabla F(\boldsymbol{x}_t)\rangle \geq \frac{1}{\beta}\|\nabla F(\boldsymbol{x}_t)\|^2.$$

Take linear combination of above inequalities,

$$\mathbb{E}_{\boldsymbol{z}_{t-1}}[\langle \boldsymbol{x}_t - \boldsymbol{x}_*, \nabla F(\boldsymbol{x}_t)\rangle] \geq \theta \nu_t \mathbb{E}_{\boldsymbol{z}_{t-1}}[\|\boldsymbol{x}_t - \boldsymbol{x}_*\|^2] + \frac{(1-\theta)}{\beta}\mathbb{E}_{\boldsymbol{z}_{t-1}}[\|\nabla F(\boldsymbol{x}_t)\|^2]$$
$$\geq \theta \nu \mathbb{E}_{\boldsymbol{z}_{t-1}}[\|\boldsymbol{x}_t - \boldsymbol{x}_*\|^2] + \frac{(1-\theta)}{\beta}\mathbb{E}_{\boldsymbol{z}_{t-1}}[\|\nabla F(\boldsymbol{x}_t)\|^2]. \tag{2}$$

Let $r_t = \|\boldsymbol{x}_t - \boldsymbol{x}_*\|$ be the solution error at step $t$. We have the following inequalities between $r_{t+1}$ and $r_t$.

$$r_{t+1}^2 = \|\boldsymbol{x}_t - \eta \nabla F(\boldsymbol{x}_t) - \eta \boldsymbol{z}_t - \boldsymbol{x}_*\|^2,$$
$$= \|\boldsymbol{x}_t - \boldsymbol{x}_*\|^2 - 2\eta \langle \nabla F(\boldsymbol{x}_t) + \boldsymbol{z}_t, \boldsymbol{x}_t - \boldsymbol{x}_*\rangle + \eta^2 \|\nabla F(\boldsymbol{x}_t) + \boldsymbol{z}_t\|^2. \tag{3}$$

Take expectation with respect to $\boldsymbol{z}_t$, we have

$$\mathbb{E}_{\boldsymbol{z}_t}[r_{t+1}^2] \leq \|\boldsymbol{x}_t - \boldsymbol{x}_*\|^2 - 2\eta \langle \nabla F(\boldsymbol{x}_t), \boldsymbol{x}_t - \boldsymbol{x}_*\rangle + \eta^2 \|\nabla F(\boldsymbol{x}_t)\|^2 + p\eta^2 \sigma_t^2. \tag{4}$$

Further take expectation with respect to $\boldsymbol{z}_{t-1}$ and use Eq 2, we have

$$\mathbb{E}_{\boldsymbol{z}_t, \boldsymbol{z}_{t-1}}[r_{t+1}^2] \leq \mathbb{E}_{\boldsymbol{z}_{t-1}}[\|\boldsymbol{x}_t - \boldsymbol{x}_*\|^2] - 2\eta \mathbb{E}_{\boldsymbol{z}_{t-1}}[\langle \nabla F(\boldsymbol{x}_t), \boldsymbol{x}_t - \boldsymbol{x}_*\rangle] + \eta^2 \mathbb{E}_{\boldsymbol{z}_{t-1}}[\|\nabla F(\boldsymbol{x}_t)\|^2] + p\eta^2 \sigma_t^2,$$
$$\leq (1 - 2(1-\theta)\eta\nu)\mathbb{E}_{\boldsymbol{z}_{t-1}}[r_t^2] + \left(\eta^2 - \frac{2\eta\theta}{\beta}\right)\mathbb{E}_{\boldsymbol{z}_{t-1}}[\|\nabla F(\boldsymbol{x}_t)\|^2] + p\eta^2 \sigma_t^2. \tag{5}$$

Set $\theta = \frac{1}{2}$ and $\eta \leq \frac{1}{\beta}$,

$$\mathbb{E}_{\boldsymbol{z}_t, \boldsymbol{z}_{t-1}}[r_{t+1}^2] \leq (1 - \eta\nu)\mathbb{E}_{\boldsymbol{z}_{t-1}}[r_t^2] + p\eta^2 \sigma_t^2. \tag{6}$$

Applying Eq (6) and taking expectation with respect to $\boldsymbol{z}_t, \boldsymbol{z}_{t-1}, \cdots, \boldsymbol{z}_1$ iteratively yields

$$\mathbb{E}[r_{t+1}^2] \leq (1 - \eta\nu)^t r_1^2 + p\eta^2 \sum_{i=1}^t (1 - \eta\nu)^{t-i} \sigma_i^2. \tag{7}$$

Uniform privacy budget allocation scheme sets

$$\sigma_t^2 = \Theta\left(\frac{TL^2 \log(1/\delta)}{n^2 \epsilon^2}\right).$$

Therefore

$$\mathbb{E}[r_{T+1}^2] \leq (1 - \eta\nu)^T r_1^2 + \Theta\left(\frac{p\eta TL^2 \log(1/\delta)}{\nu n^2 \epsilon^2}\right). \tag{8}$$

Set $T \geq \frac{2\log(n)}{\eta\nu}$, we have

$$(1 - \eta\nu)^T r_1^2 = \exp\left(\frac{\log(1 - \eta\nu)\log(n^2)}{\eta\nu}\right) r_1^2 = \exp\left(\log(1/n^2)\frac{1}{\eta\nu}\log(1 + \frac{\eta\nu}{1 - \eta\nu})\right) r_1^2,$$
$$\leq \left(\frac{1}{n^2}\right)^{\frac{1}{\eta\nu}\log(1 + \frac{\eta\nu}{1 - \eta\nu})} r_1^2 < \frac{r_1^2}{n^2}. \tag{9}$$

Last inequality holds because $\frac{1}{\eta\nu}\log(1 + \frac{\eta\nu}{1 - \eta\nu}) > 1$ for $\frac{1}{\eta\nu} \geq \frac{\beta}{\nu} \geq 1$.

Therefore, for $T \geq \frac{2 \log(n)}{\eta \nu}$, we have the excepted solution error $\mathbb{E}[r_{T+1}^2]$ satisfies

$$\mathbb{E}[r_{T+1}^2] = \mathcal{O}\left(\frac{p\eta T L^2 \log(1/\delta)}{\nu n^2 \epsilon^2}\right). \tag{10}$$

Since $F(\boldsymbol{x})$ is $\beta$-smooth, we have

$$F(\boldsymbol{x}) - F(\boldsymbol{x}_*) \leq \frac{\beta}{2}\|\boldsymbol{x} - \boldsymbol{x}_*\|^2. \tag{11}$$

Using Eq (10) and Eq (11), we have the excepted excess risk satisfies

$$\mathbb{E}[F(\boldsymbol{x}_{T+1}) - F(\boldsymbol{x}_*)] = \mathcal{O}\left(\frac{\beta p \eta T L^2 \log(1/\delta)}{\nu n^2 \epsilon^2}\right)$$

for $T \geq \frac{2 \log(n)}{\eta \nu}$. The utility bound is minimized when $T = \frac{2 \log(n)}{\eta \nu}$. $\square$

*Proof of Theorem 2.* The smooth condition gives us,

$$F(\boldsymbol{x}_{t+1}) \leq F(\boldsymbol{x}_t) + \langle \nabla F(\boldsymbol{x}_t), \boldsymbol{x}_{t+1} - \boldsymbol{x}_t \rangle + \frac{\beta}{2}\|\boldsymbol{x}_{t+1} - \boldsymbol{x}_t\|^2$$

$$= F(\boldsymbol{x}_t) - \eta \langle \nabla F(\boldsymbol{x}_t), \nabla F(\boldsymbol{x}_t) + \boldsymbol{z}_t \rangle + \frac{\beta \eta^2}{2}\|\nabla F(\boldsymbol{x}_t) + \boldsymbol{z}_t\|^2. \tag{12}$$

Take expectation with respect to $\boldsymbol{z}_t$ and substitute $\eta = \frac{1}{\beta}$,

$$\mathbb{E}_{\boldsymbol{z}_t}[F(\boldsymbol{x}_{t+1})] = F(\boldsymbol{x}_t) - \frac{1}{2\beta}\|\nabla F(\boldsymbol{x}_t)\|^2 + \frac{1}{2\beta}p\sigma_t^2. \tag{13}$$

Subtract $F(\boldsymbol{x}_*)$ on both sides and use convexity,

$$\mathbb{E}_{\boldsymbol{z}_t}[F(\boldsymbol{x}_{t+1}) - F(\boldsymbol{x}_*)] = F(\boldsymbol{x}_t) - F(\boldsymbol{x}_*) - \frac{1}{2\beta}\|\nabla F(\boldsymbol{x}_t)\|^2 + \frac{1}{2\beta}p\sigma_t^2$$

$$\leq \langle \nabla F(\boldsymbol{x}_t), \boldsymbol{x}_t - \boldsymbol{x}_* \rangle - \frac{1}{2\beta}\|\nabla F(\boldsymbol{x}_t)\|^2 + \frac{1}{2\beta}p\sigma_t^2. \tag{14}$$

Substitute $\nabla F(\boldsymbol{x}_t) = \beta(\boldsymbol{x}_t - \boldsymbol{x}_{t+1}) - \boldsymbol{z}_t$,

$$\mathbb{E}_{\boldsymbol{z}_t}[F(\boldsymbol{x}_{t+1}) - F(\boldsymbol{x}_*)] \leq \beta\langle \boldsymbol{x}_t - \boldsymbol{x}_{t+1}, \boldsymbol{x}_t - \boldsymbol{x}_* \rangle - \frac{1}{2\beta}\mathbb{E}_{\boldsymbol{z}_t}[\|\beta(\boldsymbol{x}_t - \boldsymbol{x}_{t+1}) - \boldsymbol{z}_t\|^2] + \frac{1}{2\beta}p\sigma_t^2$$

$$= \beta\langle \boldsymbol{x}_t - \boldsymbol{x}_{t+1}, \boldsymbol{x}_t - \boldsymbol{x}_* \rangle - \frac{\beta}{2}\|\boldsymbol{x}_t - \boldsymbol{x}_{t+1}\|^2 - \mathbb{E}_{\boldsymbol{z}_t}\langle \boldsymbol{x}_{t+1}, \boldsymbol{z}_t \rangle$$

$$= \beta\langle \boldsymbol{x}_t - \boldsymbol{x}_{t+1}, \boldsymbol{x}_t - \boldsymbol{x}_* \rangle - \frac{\beta}{2}\|\boldsymbol{x}_t - \boldsymbol{x}_{t+1}\|^2 - \mathbb{E}_{\boldsymbol{z}_t}\langle \boldsymbol{x}_t - \eta\nabla F(\boldsymbol{x}_t) - \eta\boldsymbol{z}_t, \boldsymbol{z}_t \rangle$$

$$= \beta\langle \boldsymbol{x}_t - \boldsymbol{x}_{t+1}, \boldsymbol{x}_t - \boldsymbol{x}_* \rangle - \frac{\beta}{2}\|\boldsymbol{x}_t - \boldsymbol{x}_{t+1}\|^2 + \frac{1}{\beta}p\sigma_t^2$$

$$= \frac{\beta}{2}(\|\boldsymbol{x}_t - \boldsymbol{x}_*\|^2 - \|\boldsymbol{x}_{t+1} - \boldsymbol{x}_*\|^2) + \frac{1}{\beta}p\sigma_t^2. \tag{15}$$

Summing over $t = 1, \ldots, T$ and take expectation with respect to $\boldsymbol{z}_1, \ldots, \boldsymbol{z}_T$,

$$\sum_{t=1}^{T}\mathbb{E}[F(\boldsymbol{x}_{t+1}) - F(\boldsymbol{x}_*)] \leq \frac{\beta}{2}\|\boldsymbol{x}_1 - \boldsymbol{x}_*\|^2 + \sum_{t=1}^{T}\frac{1}{\beta}p\sigma_t^2. \tag{16}$$

Use convexity,

$$\mathbb{E}[F(\bar{\boldsymbol{x}}) - F(\boldsymbol{x}_*)] \leq \frac{\beta}{2T}\|\boldsymbol{x}_1 - \boldsymbol{x}_*\|^2 + \frac{1}{\beta}p\sigma^2$$

$$\leq \frac{\beta}{2T}\|\boldsymbol{x}_1 - \boldsymbol{x}_*\|^2 + \Theta\left(\frac{L^2 p T \log(1/\delta)}{\beta n^2 \epsilon^2}\right) \tag{17}$$

Choose $T = \frac{n\beta\epsilon}{\sqrt{p}}$, we have

$$\mathbb{E}[F(\bar{\boldsymbol{x}}) - F(\boldsymbol{x}_*)] = \mathcal{O}\left(\frac{\sqrt{p}L^2 \log(1/\delta)}{n\epsilon}\right). \tag{18}$$

$\square$

*Proof of Theorem 3 and 4.* We start by giving a useful lemma.

**Lemma 2.** *Choose* $\eta_t = \frac{1}{\nu t}$, *the expected solution error of* $\boldsymbol{x}_t$ *in Algorithm 2 for any* $t > 1$ *satisfies*

$$\mathbb{E}[\|\boldsymbol{x}_t - \boldsymbol{x}_*\|^2] \leq \frac{2L^2 \left(1 + p\sigma^2\right)}{t\nu^2},$$

*Proof of Lemma 2.* We have

$$
\begin{aligned}
\|\boldsymbol{x}_{t+1} - \boldsymbol{x}_*\|^2 &= \|\boldsymbol{x}_t - \eta_t \boldsymbol{g}_t - \eta_t \boldsymbol{z}_t - \boldsymbol{x}_*\|^2 \\
&= \|\boldsymbol{x}_t - \boldsymbol{x}_*\|^2 - 2\eta_t \langle \boldsymbol{x}_t - \boldsymbol{x}_*, \boldsymbol{g}_t + \boldsymbol{z}_t \rangle + \eta_t^2 \|\boldsymbol{g}_t\|^2 - 2\eta_t^2 \langle \boldsymbol{g}_t, \boldsymbol{z}_t \rangle + \eta_t^2 \|\boldsymbol{z}_t\|^2 .
\end{aligned}
\tag{19}
$$

Take expectation with respect to perturbation noise $\boldsymbol{z}_t$ and uniform sampling, we have

$$
\begin{aligned}
\mathbb{E}_{\boldsymbol{z}_t, i_t}[\|\boldsymbol{x}_{t+1} - \boldsymbol{x}_*\|^2] &= \mathbb{E}_{\boldsymbol{z}_t, i_t}[\|\boldsymbol{x}_t - \eta_t \boldsymbol{g}_t - \eta_t \boldsymbol{z}_t - \boldsymbol{x}_*\|^2] \\
&\leq \|\boldsymbol{x}_t - \boldsymbol{x}_*\|^2 - 2\eta_t \langle \boldsymbol{x}_t - \boldsymbol{x}_*, \nabla F\left(\boldsymbol{x}_t\right)\rangle + \eta_t^2 L^2 + p\eta_t^2 L^2 \sigma^2.
\end{aligned}
\tag{20}
$$

Further take expectation to $\boldsymbol{z}_{t-1}$ and apply Definition 3,

$$
\begin{aligned}
\mathbb{E}_{\boldsymbol{z}_t, \boldsymbol{z}_{t-1}, i_t}[\|\boldsymbol{x}_{t+1} - \boldsymbol{x}_*\|^2] &\leq \left(1 - 2\nu_t \eta_t\right) \mathbb{E}_{\boldsymbol{z}_{t-1}}[\|\boldsymbol{x}_t - \boldsymbol{x}_*\|^2] + \eta_t^2 L^2 \left(1 + p\sigma^2\right) \\
&\leq \left(1 - 2\nu \eta_t\right) \mathbb{E}_{\boldsymbol{z}_{t-1}}[\|\boldsymbol{x}_t - \boldsymbol{x}_*\|^2] + \eta_t^2 L^2 \left(1 + p\sigma^2\right).
\end{aligned}
\tag{21}
$$

Now we use induction to conduct the proof. Substitute $\eta_t = \frac{1}{t\nu}$ into Eq 21, we have Lemma 2 hold for $t = 2$.

Assume $\mathbb{E}[\|\boldsymbol{x}_t - \boldsymbol{x}_*\|^2] \leq \frac{2L^2 \left(1 + p\sigma^2\right)}{t\nu^2}$ holds for $t > 2$, then

$$
\begin{aligned}
\mathbb{E}[\|\boldsymbol{x}_{t+1} - \boldsymbol{x}_*\|^2] &\leq \left(1 - \frac{2}{t}\right) \mathbb{E}[\|\boldsymbol{x}_t - \boldsymbol{x}_*\|^2] + \frac{L^2 \left(1 + p\sigma^2\right)}{\nu^2 t^2} \\
&\leq \left(\frac{1}{t} - \frac{2}{t^2}\right) \frac{2L^2 \left(1 + p\sigma^2\right)}{\nu^2} + \frac{L^2 \left(1 + p\sigma^2\right)}{\nu^2 t^2} \\
&= \left(\frac{2}{t} - \frac{3}{t^2}\right) \frac{L^2 \left(1 + p\sigma^2\right)}{\nu^2} \leq \frac{2L^2 \left(1 + p\sigma^2\right)}{\left(t+1\right) \nu^2}.
\end{aligned}
\tag{22}
$$

$\square$

It's easy to check that Eq 20 holds for arbitrary $\boldsymbol{x}$ rather than $\boldsymbol{x}_*$. Rearrange Eq 20 and take expectation, we have

$$\mathbb{E}[\langle \boldsymbol{x}_t - \boldsymbol{x}, \nabla F\left(\boldsymbol{x}_t\right)\rangle] \leq \frac{\mathbb{E}[\|\boldsymbol{x}_t - \boldsymbol{x}\|^2] - \mathbb{E}[\|\boldsymbol{x}_{t+1} - \boldsymbol{x}\|^2]}{2\eta_t} + \frac{\eta_t L^2 \left(1 + p\sigma^2\right)}{2}.
\tag{23}$$

Let $k$ be arbitrarily chosen from $\{1, \ldots, \lfloor T/2 \rfloor\}$. Summing over the last $k + 1$ iterations and use convexity to lower bound $\langle \boldsymbol{x}_t - \boldsymbol{x}, \nabla F\left(\boldsymbol{x}_t\right)\rangle$ by $F\left(\boldsymbol{x}_t\right) - F\left(\boldsymbol{x}\right)$,

$$
\begin{aligned}
\sum_{t=T-k}^{T} \mathbb{E}[F\left(\boldsymbol{x}_t\right) - F\left(\boldsymbol{x}\right)] \leq {} & \frac{\mathbb{E}[\|\boldsymbol{x}_{T-k} - \boldsymbol{x}\|^2]}{2\eta_{T-k}} + \frac{1}{2} \sum_{t=T-k+1}^{T} \mathbb{E}[\|\boldsymbol{x}_t - \boldsymbol{x}\|^2] \left(\frac{1}{n_t} - \frac{1}{n_{t-1}}\right) \\
& - \frac{\mathbb{E}[\|\boldsymbol{x}_{T+1} - \boldsymbol{x}\|^2]}{2\eta_T} + \frac{L^2 \left(1 + p\sigma^2\right)}{2} \sum_{t=T-k}^{T} \eta_t.
\end{aligned}
\tag{24}
$$

Substitute $\eta_t = \frac{1}{\nu t}$ and follow the idea in Shamir & Zhang (2013) by choosing $\boldsymbol{x} = \boldsymbol{x}_{T-k}$, we arrive at

$$\sum_{t=T-k}^{T} \mathbb{E}[F\left(\boldsymbol{x}_t\right) - F\left(\boldsymbol{x}_{T-k}\right)] \leq \frac{\nu}{2} \sum_{t=T-k+1}^{T} \mathbb{E}[\|\boldsymbol{x}_t - \boldsymbol{x}_{T-k}\|^2] + \frac{L^2 \left(1 + p\sigma^2\right)}{2\nu} \sum_{t=T-k}^{T} \frac{1}{t}.
\tag{25}$$

Now we bound $\mathbb{E}[\|\boldsymbol{x}_t - \boldsymbol{x}_{T-k}\|^2]$ for $t \geq T - k$,

$$
\begin{aligned}
\mathbb{E}[\|\boldsymbol{x}_t - \boldsymbol{x}_{T-k}\|^2] &\leq 2\mathbb{E}[\|\boldsymbol{x}_t - \boldsymbol{x}_*\|^2] + 2\mathbb{E}[\|\boldsymbol{x}_{T-k} - \boldsymbol{x}_*\|^2] \\
&\leq \frac{4L^2\left(1 + p\sigma^2\right)}{\nu^2}\left(\frac{1}{t} + \frac{1}{T-k}\right) \leq \frac{8L^2\left(1 + p\sigma^2\right)}{\nu^2}\left(\frac{1}{T-k}\right) \\
&\leq \frac{16L^2\left(1 + p\sigma^2\right)}{T\nu^2}.
\end{aligned}
\tag{26}
$$

Substitute Eq 26 into Eq 25,

$$
\sum_{t=T-k}^{T} \mathbb{E}[F\left(\boldsymbol{x}_t\right) - F\left(\boldsymbol{x}_{T-k}\right)] \leq \frac{8kL^2\left(1 + p\sigma^2\right)}{T\nu} + \frac{L^2\left(1 + p\sigma^2\right)}{2\nu} \sum_{t=T-k}^{T} \frac{1}{t}.
\tag{27}
$$

Let $S_k = \frac{1}{k+1}\sum_{t=T-k}^{T} \mathbb{E}[F\left(\boldsymbol{x}_t\right)]$ be the averaged expected values of the last $k+1$ iterations. We are interested in $S_0 - F\left(\boldsymbol{x}_*\right) = \mathbb{E}[F\left(\boldsymbol{x}_T\right)] - F\left(\boldsymbol{x}_*\right)$. Now we derive an inequality between $S_k$ and $S_{k-1}$. By definition,

$$
kS_{k-1} = (k+1)S_k - \mathbb{E}[\boldsymbol{x}_{T-k}].
\tag{28}
$$

Rearrange Eq 27 to upper bound $-\mathbb{E}[\boldsymbol{x}_{T-k}]$,

$$
\begin{aligned}
S_{k-1} &= \frac{k+1}{k}S_k - \frac{\mathbb{E}[\boldsymbol{x}_{T-k}]}{k} \\
&\leq \frac{k+1}{k}S_k - \frac{S_k}{k} + \frac{8L^2\left(1 + p\sigma^2\right)}{(k+1)T\nu} + \frac{L^2\left(1 + p\sigma^2\right)}{2k(k+1)\nu} \sum_{t=T-k}^{T} \frac{1}{t} \\
&\leq S_k + \frac{L^2\left(1 + p\sigma^2\right)}{2\nu}\left(\frac{16}{kT} + \frac{1}{k(k+1)} \sum_{t=T-k}^{T} \frac{1}{t}\right).
\end{aligned}
\tag{29}
$$

Summing over $k = 1, \ldots, k = \lfloor T/2 \rfloor$,

$$
S_0 \leq S_{\lfloor T/2 \rfloor} + \frac{L^2\left(1 + p\sigma^2\right)}{2\nu}\left(\sum_{k=1}^{\lfloor T/2 \rfloor} \frac{16}{kT} + \sum_{k=1}^{\lfloor T/2 \rfloor} \sum_{t=T-k}^{T} \frac{1}{k(k+1)t}\right).
\tag{30}
$$

Now we bound $S_{\lfloor T/2 \rfloor} - F(\boldsymbol{x}_*)$. Choose $\boldsymbol{x} = \boldsymbol{x}_*$ and $\eta_t = \frac{1}{t\nu}$ in Eq 24 ,

$$
\begin{aligned}
\sum_{t=\lceil T/2 \rceil}^{T} \mathbb{E}[F\left(\boldsymbol{x}_t\right) - F\left(\boldsymbol{x}_*\right)] &= \frac{\nu\lceil T/2 \rceil\mathbb{E}[\|\boldsymbol{x}_{\lceil T/2 \rceil} - \boldsymbol{x}_*\|^2]}{2} + \frac{\nu}{2} \sum_{t=\lceil T/2 \rceil+1}^{T} \mathbb{E}[\|\boldsymbol{x}_t - \boldsymbol{x}_*\|^2] \\
&\quad + \frac{L^2\left(1 + p\sigma^2\right)}{2} \sum_{t=\lceil T/2 \rceil}^{T} \eta_t \\
&\leq \frac{L^2(1 + p\sigma^2)}{\nu}(1 + \sum_{t=\lceil T/2 \rceil+1}^{T} \frac{1}{t} + \sum_{t=\lceil T/2 \rceil}^{T} \frac{1}{2t}) \\
&\leq \frac{L^2(1 + p\sigma^2)}{\nu}(1 + \frac{3}{2} \sum_{t=\lceil T/2 \rceil}^{T} \frac{1}{t}) \\
&\leq \frac{4L^2(1 + p\sigma^2)}{\nu}.
\end{aligned}
\tag{31}
$$

The second inequality uses Lemma 2. The last inequality holds because the fact that $\sum_{t=\lceil T/2 \rceil}^{T} \frac{1}{t} \leq \log(2)$. Dividing Eq 31 by $\lceil T/2 \rceil$,

$$
S_{\lfloor T/2 \rfloor} - F(\boldsymbol{x}_*) \leq \frac{8L^2(1 + p\sigma^2)}{T\nu}.
\tag{32}
$$

We have $\sum_{k=1}^{\lfloor T/2 \rfloor} \frac{16}{kT} \leq \frac{16(1+log(T))}{T}$ because it is harmonic sequence. Lastly,

$$\sum_{k=1}^{\lfloor T/2 \rfloor} \sum_{t=T-k}^{T} \frac{1}{k\,(k+1)\,t} \leq \sum_{k=1}^{\lfloor T/2 \rfloor} \frac{\log(2)}{k(k+1)}$$
$$\leq \sum_{k=1}^{\lfloor T/2 \rfloor} \frac{\log(2)}{k^2} \leq 2\log(2).$$

(33)

Plugging these bounds into Eq 30, we have

$$S_0 - F(\boldsymbol{x}_*) = \mathcal{O}\left( \frac{(1+p\sigma^2)L^2 \log(T)}{T\nu} \right).$$

(34)

Choose $\sigma^2 = \Theta\left( \frac{T log(1/\delta)}{n^2 \epsilon^2} \right)$ to guarantee $(\epsilon, \delta)$-DP. Set $T = n^2 \epsilon^2$, we have

$$S_0 - F(\boldsymbol{x}_*) = \mathcal{O}\left( \frac{pL^2 \log(n) log\,(1/\delta)}{n^2 \epsilon^2 \nu} \right).$$

(35)

Set $T = \frac{n\epsilon}{\sqrt{p}}$ and assume $p < n^2$, we have

$$S_0 - F(\boldsymbol{x}_*) = \mathcal{O}\left( \frac{\sqrt{p}L^2 \log(n)}{n\epsilon\nu} \right).$$

(36)

$\square$

## APPENDIX B   DETAILED DESCRIPTION ON BENCHMARK DATASETS

Table 3: Detailed description of seven real world datasets.

| dataset | Adult | KDDCup99 | MNIST | Covertype | Gisette | Real-sim | RCV1 |
|---|---|---|---|---|---|---|---|
| # records | 45220 | 70000 | 65000 | 581012 | 6000 | 72309 | 50000 |
| # features | 104 | 114 | 784 | 54 | 5000 | 20958 | 47236 |
| # classes | 2 | 2 | 10 | 7 | 2 | 2 | 2 |

## APPENDIX C   COMPARISON BETWEEN AVERAGE AND MINIMUM CURVATURES ON DIFFERENT DATASET

In this section we plot the average and minimum curvatures in Figure 4 for another dataset KDDCup99. The objective function is still regularized logistic regression.

As shown in Figure 4, the average curvature is still larger than the minimum curvature (especially when the regularization term is small). Despite this, the average curvature of KDDCup99 is smaller than Adult, this may be the reason why the improvement in Section 4 is larger for the Adult dataset.

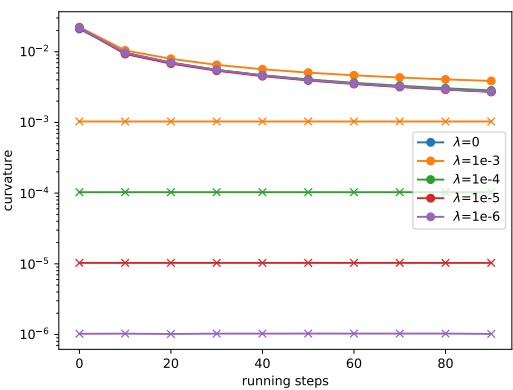

Figure 4: Curvatures of regularized logistic regression on KDDCup99 dataset over training. Dot symbol represents average curvature and cross symbol represents minimum curvature.

