# OpenReview forum: "Gradient Perturbation is Underrated for Differentially Private Convex Optimization"
_ICLR.cc/2020/Conference — Reject_

### Official Review · AnonReviewer3 · 2019-10-23
**Official Blind Review #3**

**Rating:** 6

**Review:**

Differential privacy (DP) can be achieved by perturbing the objective function, the output or the gradient. In this paper, the authors consider gradient perturbation and claim that it is more advantageous than other methods. To prove this claim, they present a novel utility analysis by taking the noise into account. The previous papers (like Bassily et. al) present utility guarantees in DP context, but their analysis follow the same steps used for non-noisy setting. In non-noisy setting, the analysis is based on strong convexity parameter \mu which is the minimum curvature. However, in this study they present ”expected curvature” which is computed by considering the noise variance and based on averaging the curvatures along the number of iterations. The order of utility is given for both convex and strongly convex objectives and it has become smaller than the previous studies. Since other perturbation methods does not add noise at intermediate steps, expected curvature is the same with \mu and the utility advantage is not valid.

Comments (Positive)

- The paper is well-written and easy to follow. I didn’t see typos or mistakes (I didn’t check the last proof).

- They claim that they are the first study showing the advantage of gradient perturbation theoretically (I haven’t seen such a study either).

- Since they remove the dependency to minimum curvature \mu, they present utility order for both convex and strongly convex objective for DP-GD and DP-SGD.

- With the help of privacy noise, they obtain a better utility which is an interesting contribution.

Comments (Negative)

- In the numerical experiments, number of iterations are taken as 20, 200 and 800. It might be checked for more iterations. The chosen privacy levels are tight enough (0.1 - 1).

- The learning rate of DP-SGD is divided by 2 at the middle of training. The reason and whether it is applied to other SGD method is not clear.


Overall, this type of utility analysis exists in the DP literature, but their novelty comes from the idea of averaging the curvature.

**Experience Assessment:**

I have published one or two papers in this area.

**Review Assessment: Checking Correctness Of Derivations And Theory:**

I assessed the sensibility of the derivations and theory.

**Review Assessment: Checking Correctness Of Experiments:**

I carefully checked the experiments.

**Review Assessment: Thoroughness In Paper Reading:**

I read the paper at least twice and used my best judgement in assessing the paper.

---

> ### Author Response · Authors · 2019-11-09
> **Response to Reviewer #3**
>
> Thank you for your recognition of our work!  We hope the below responses can address your concerns.
>
> 1.  Regards to running steps, we found the given three steps are enough for DP-SGD and DP-GD to achieve good performance. We did test more settings on low-dimensional datasets and found there is no significant improvement. For baseline algorithms, we use the hyperparameters in Iyengar et al, which are tuned thoroughly.
> As for the chosen privacy levels, 0.1~1.0 may be tight for recent deep learning models (i.e. the settings in Abadi et al. (2016)), but they are reasonable for small datasets and simple objectives. For larger privacy parameter (i.e. epsilon=5), the benefit of DP-SGD and DP-GD becomes smaller as the performance of all algorithms are similar to the non-private one.
>
> 2.  The learning rate of SGD with output perturbation (Out-SGD) is also decreased during training. However, the schedule is not the same as ours. Out-SGD decreases the learning rate with respect to current running step. Use “constant-and-cut” schedule would yield larger sensitivity (therefore larger noise) for Out-SGD. See the comparison between Corollary 1&2 as well as Lemma 7&8 in [1] for the difference between two schedules.
>
> [1]: Xi Wu, Fengan Li, Arun Kumar, Kamalika Chaudhuri, Somesh Jha, and Jeffrey Naughton. Bolt-on differential privacy for scalable stochastic gradient descent-based analytics. In ACM International Conference on Management of Data, 2017.

---

### Official Review · AnonReviewer4 · 2019-10-25
**Official Blind Review #4**

**Rating:** 3

**Review:**

This paper studies the problem of differentially private optimization in the (strongly) convex setting. The authors focus on the gradient perturbation methods, i.e., DP-GD and DP-SGD, and provide the utility guarantees of DP-GD and DP-SGD under the so called expected curvature assumption. However, it is very hard to verify the expected curvature assumption, and thus the theoretical results may be invalid. I summarize my main concerns as follows:

1. All the theoretical results provided in this paper are based on the expected curvature assumption (Definition 3). However, it is unclear what kind of loss functions will satisfy this assumption. If the authors can prove that this assumption can hold for some specific loss functions, such as logistic loss or square loss, the contributions of the current paper will be much stronger.
2. Again it is unclear how large $\nu$ will be compared to $\mu$, and thus the theoretical results can be useless.
3. Since the authors use the approximation form of the gradient in equation (2), why the first inequality in equation (2) holds according to Definition 3?
4. How do you get the average and minimum curvatures in Figure 1?
5. I don’t think the argument in the last paragraph in section 3.1 is sound. Because for the restricted strongly convex function, we will ensure the loss curvatures stay in certain directions during the training process.
6. The contribution of the current paper is very incremental. All the proofs are just replacing the strongly convex condition with the expected curvature condition.
7. There are several gradient perturbation based DP algorithms [1,2] for solving high-dimensional problems are missing in the related work.

Reference:
[1].Talwar, Kunal, Abhradeep Guha Thakurta, and Li Zhang. "Nearly optimal private lasso." Advances in Neural Information Processing Systems. 2015.
[2].Wang, Lingxiao, and Quanquan Gu. "Differentially Private Iterative Gradient Hard Thresholding for Sparse Learning." 28th International Joint Conference on Artificial Intelligence. 2019.

**Experience Assessment:**

I have published one or two papers in this area.

**Review Assessment: Checking Correctness Of Derivations And Theory:**

I carefully checked the derivations and theory.

**Review Assessment: Checking Correctness Of Experiments:**

I carefully checked the experiments.

**Review Assessment: Thoroughness In Paper Reading:**

I read the paper thoroughly.

---

> ### Author Response · Authors · 2019-11-09
> **Response to Reviewer #4**
>
> Thank you for your effort and review. Our responses are listed as below.
>
> 1&2.  We think there is a misunderstanding about expected curvature. The expected curvature is not assumed but naturally exists. We introduce $\nu$ to denote the expected curvature in Definition (3). We compare the value of $\nu$ and the usual minimum curvature $\mu$ in the paragraph “In fact, $\nu$ represents the average curvature and is much larger than $\mu$”. Approximately, we have $\nu\approx tr(H_x)/p$ while $\mu$ is the smallest eigenvalue of $H_x$.
>
>
> 3.  There is a misunderstanding. The equation (2) is not derived from Definition (3). In equation (2), we just approximate the left hand side by second order Taylor expansion and express out the right hand side, respectively. The approximation is reasonably accurate for smooth convex objectives. We then estimate the value of $\nu$ in the paragraph “In fact, $\nu$ represents the average curvature and is much larger than $\mu$”. We rewrite the equation (2) into separated formulas for left and right hand sides in the updated paper.
>
>
> 4.  We use the API provided by deep learning framework Tensorflow. More specifically, we first use tf.hessians to compute the Hessian matrix at each step. Then we use tf.linalg.eigh to compute its eigenvalues (the curvatures). Though we use GPU accelerator, the computation cost of high-dimensional dataset is still unacceptable. Therefore, we choose one of the low-dimensional datasets. We are happy to release our source code after the review period.
> https://www.tensorflow.org/api_docs/python/tf/hessians
> https://www.tensorflow.org/api_docs/python/tf/linalg/eigh
>
>
> 5.  We use Figure 2 as an example to show that $\mu=0$ does not necessary lead to expected curvature $\nu=0$.  We agree that there exists analysis strategy to avoid using $\mu$ to derive the utility guarantee in such case and we add reference (the missed one [2]) in the updated paper to clarify this.
>
> 6.  We give the new utility bounds of the gradient perturbation approach for the empirical minimization problem with differential privacy guarantee. We analyzed both DP-GD and DP-SGD for both convex and strongly convex objectives. In the analysis we point out that the noise added for guaranteeing privacy can help the optimization analysis to obtain better utility guarantee for gradient perturbation approach. Our new utility bound correctly reflect the practically observed advantage of gradient perturbation over output/objective perturbation, which is new and novel for understanding the DP-ERM problem. We hope the reviewer can reevaluate our contribution.
>
> 7.  We have reference to [1] in the second paragraph of introduction in our original paper. We missed the recently published paper [2]. In the updated paper, we add the reference [2] and discussion in Section 3.1.

---

### Official Review · AnonReviewer2 · 2019-10-25
**Official Blind Review #2**

**Rating:** 6

**Review:**

This paper proposes a quantity called expected curvature to analyze the convergence of gradient perturbation based methods that achieves differential privacy. Comparing to minimum curvature, which was used in previous convergence analyses, expected curvature better captures the properties of the optimization problem, and thus offers an explanation for the advantage of gradient perturbation based methods over objective perturbation and output perturbation.
Using expected curvature is a pretty interesting idea, and having more refined convergence bound is useful. I have the following questions.
1. It seems to me that the convergence bound is similar to previous bound, except that \mu is replaced by \nu and one log(n) disappears. Could you explain more intuitively how hard the new analysis is? Is it similar to just replacing any \mu by \nu in the previous analysis? And why do they differ by log(n)?
2. How are the experiments (in terms of setup and results) differ from those in Iyengar et al? It seems to me the paper is proposing a method for convergence analysis and the DP algorithms remain the same. I feel like in Iyengar et al, there is no clear difference between gradient perturbation and objective perturbation. Maybe I was wrong about that, but could you elaborate more?
3. I agree that the expected curvature better captures the convergence of gradient-based DP methods. Yet I don’t see clearly how this can be used to show that they have more advantages than objective perturbation. Is it possible that the analyses of objective perturbation can also be improved (maybe with other techniques)? Since all we have are upper bounds, I feel like it is a bit early to conclude that it is less powerful. You mentioned that “That is because DP makes the worst-case assumption on query function and output/objective perturbation treat the whole learning algorithm as a single query to private dataset. ” I didn’t follow this part. I feel like DP is always making a worst-case assumption; even for gradient perturbation, you need to add noise to protect the worst case. Could you elaborate more on that?
4. My understanding is that for different dataset the curvatures would be different. I think it might be interesting if you plot something similar to Figure 1 for other datasets and compare how they match with the training curve. Do you expect them to look very different on different dataset / optimization problem?

**Experience Assessment:**

I have published one or two papers in this area.

**Review Assessment: Checking Correctness Of Derivations And Theory:**

I assessed the sensibility of the derivations and theory.

**Review Assessment: Checking Correctness Of Experiments:**

I assessed the sensibility of the experiments.

**Review Assessment: Thoroughness In Paper Reading:**

I read the paper at least twice and used my best judgement in assessing the paper.

---

> ### Author Response · Authors · 2019-11-09
> **Response to Reviewer #2**
>
> Thank you for the review. We hope the following answers can address your concerns.
>
> 1.  Indeed, our convergence analysis shares the same procedure as the classical optimization analysis. Such analysis is widely used in the DP-ERM literature (i.e. Bassily et al. (2014)). But it is not simply replacing $\mu$ by $\nu$. We extract the dependence of the optimization analysis on expectation curvature $\nu$ by carefully manipulating the added noise. Moreover, we analyze the utility of DP-GD for convex objective that previous work has not covered before (Jayaraman et al. (2018) only study the strongly convex case). Nonetheless, our major contribution is not inventing new optimization technique but pointing out that the noise added for guaranteeing privacy can help the optimization analysis to obtain better utility guarantee for gradient perturbation approach. Such a benefit cannot be utilized by output/objective perturbation directly. We believe our utility bound correctly reflect the practically observed advantage of gradient perturbation over output/objective perturbation, which is new and novel for understanding the DP-ERM problem.
> For DP-SGD, we improve the bound by a log(n) factor because we use a tighter composition theorem (Abadi et al. (2016)). For DP-GD, we save a log(n) factor by relaxing the inequality (9) more carefully.
>
> 2.  For DP-SGD, our setup has two differences compared with Iyengar et al. First, Iyengar et al use an upper bound to analyze the privacy loss and therefore yields larger noise variance than actually needed (Algorithm 2 in their Appendix C). We follow Abadi et al. (2016) to compute the exact privacy loss through numerical integration. As a result, we need smaller variance than Iyengar et al. to provide the same level privacy guarantee. Second, we decay the learning rate of DP-SGD during training while Iyengar et al use constant learning rate. Learning rate decay is important for both theoretical analysis and empirical performance of DP-SGD. The above differences lead to a significant improvement on the performance of DP-SGD.
>
> 3.  Gradient perturbation outperforms other perturbation methods empirically and our tighter utility upper bounds provide theoretical justification.
>
> It is true that DP is always making a worst-case assumption; even for gradient perturbation. The major difference is that the worst-case assumption of gradient perturbation (sensitivity) is on the gradient norm, e.g., the function is L-Lipschitz, while the optimization analysis depends on the smoothness and strongly convex property. This leaves us room to consider how the added noise affects the optimization analysis. On the contrary, output/objective perturbations treat the whole learning process as a single query and the worst-case assumption (sensitivity) is on the convergence property of the learning algorithm which is directly related to the smoothness and strongly convex coefficient. This is the reason why it is hard to improve the utility analysis for the objective/output perturbations from the optimization side.
>
> 4.  The average curvature could be different for different datasets because the Hessian matrix of the loss is data dependent and the minimum curvature is usually determined by the $L_2$ regularization term when the Hessian of the loss is degenerated. We plot the  curvatures for another low-dimensional dataset KDDCup99 in the Appendix C (computing the curvatures is very costly as it requires the computation of the Hessian matrix at each step). We observe that the average curvature of KDDCup99 is still much larger than the minimum curvature, especially when the regularization term is small. We also observe that the average curvature of KDDCup99 is indeed different from that of the Adult dataset as the Hessian is data dependent.

---

### Author Response · Authors · 2019-11-10
**General Response**

We thank all the reviewers for their constructive comments. We have uploaded our response to each reviewer. We also update the paper according to the comments. The main changes are as follows.
1. In Appendix C, we plot the curvatures for another dataset KDDCup99.
2. We rewrite Equation (2) for better readability.
3. Add missing reference.

---

### Decision · Program_Chairs · 2019-12-19

**Decision:**

Reject

**Comment:**

In this paper, the authors showed that for differentially private convex optimization, the utility guarantee of both DP-GD and  DP-SGD is determined by the expected curvature rather than the worst-case minimum curvature. Based on this motivation, the authors justified the advantage of gradient perturbation over other perturbation methods. This is a borderline paper, and has been discussed after author response. The main concerns of this paper include (1) the authors failed to show any loss function that can satisfy the expected curvature inequality; (2) the contribution of this paper is limited, since all the proofs in the paper are just small tweak of existing proofs; (3) this paper does not really improve any existing gradient perturbation based differentially private methods. Due to the above concerns, I have to recommend reject.